# UV-Induced Somatic Mutations Driving Clonal Evolution in Healthy Skin, Nevus, and Cutaneous Melanoma

**DOI:** 10.3390/life12091339

**Published:** 2022-08-29

**Authors:** Alba Loras, Marta Gil-Barrachina, María Ángeles Marqués-Torrejón, Gemma Perez-Pastor, Conrado Martinez-Cadenas

**Affiliations:** 1Department of Medicine, University of Valencia, 46010 Valencia, Spain; 2Department of Medicine, Jaume I University of Castellon, 12071 Castellon, Spain; 3Department of Dermatology, Valencia General University Hospital, 46014 Valencia, Spain

**Keywords:** cutaneous melanoma, nevus, genomic mutations, clonal evolution, UV exposure, mutational signatures

## Abstract

Introduction: Due to its aggressiveness, cutaneous melanoma (CM) is responsible for most skin cancer-related deaths worldwide. The origin of CM is closely linked to the appearance of UV-induced somatic mutations in melanocytes present in normal skin or in CM precursor lesions (nevi or dysplastic nevi). In recent years, new NGS studies performed on CM tissue have increased the understanding of the genetic somatic changes underlying melanomagenesis and CM tumor progression. Methods: We reviewed the literature using all important scientific databases. All articles related to genomic mutations in CM as well as normal skin and nevi were included, in particular those related to somatic mutations produced by UV radiation. Conclusions: CM development and progression are strongly associated with exposure to UV radiation, although each melanoma subtype has different characteristic genetic alterations and evolutionary trajectories. While *BRAF* and *NRAS* mutations are common in the early stages of tumor development for most CM subtypes, changes in *CDKN2A, TP53* and *PTEN,* together with *TERT* promoter mutations, are especially common in advanced stages. Additionally, large genome duplications, loss of heterozygosity, and copy number variations are hallmarks of metastatic disease. Finally, the mutations driving melanoma targeted-therapy drug resistance are also summarized. The complete sequential stages of clonal evolution leading to CM onset from normal skin or nevi are still unknown, so further studies are needed in this field to shed light on the molecular pathways involved in CM malignant transformation and in melanoma acquired drug resistance.

## 1. Introduction

UV exposure is the most frequent environmental factor involved in skin cancer development, including cutaneous melanoma (CM). Melanoma is a malignant tumor that originates from melanocytes, a type of melanin-producing pigment cell present in non-glabrous or glabrous skin, eyes, and mucosal epithelia. Melanoma histologic localization is traditionally used to classify melanomas as cutaneous melanoma (CM), acral melanoma, uveal melanoma, or mucosal melanoma, respectively. Each of these melanoma subtypes presents different characteristic mutational genetic signatures. While the origin of CM is strongly related to ultraviolet radiation (UVR) (evidenced by the presence of several UV mutational signatures in CM tissue), acral, uveal, and mucosal melanomas have mutational signatures not associated with UVR exposure, suggesting that UVR has no crucial role in their pathogenesis [1]. This review focuses on the association between UVR and melanoma, and will thus concentrate solely on the genomic mutations triggering the origin and development of CM.

CM accounts for less than 5% of skin cancers but, due to its aggressiveness, it is responsible for most of the deaths caused by skin cancer worldwide [2]. The origin of CM is closely linked to cumulative sun damage (CSD) exerted by chronic or intermittent exposure to UVR, so these tumors have been classified in several categories according to this characteristic (high CSD and low CSD). While superficial spreading melanomas (SSM) are classified as low-CSD melanomas, lentigo maligna melanoma (LMM) fall into the group of high-CSD melanomas. Finally, the remaining common subtype of CM, nodular melanoma (NM), shows common characteristics to both the low- and the high-CSD types, since some subsets of NM are associated with low CSD, others with high CSD, and still others with neither [3].

CM is developed sequentially from the accumulation and selection of somatic mutations produced in normal skin or in nevi [4]. In fact, both nevus and especially dysplastic nevus (DN) represent an intermediate step in melanoma carcinogenesis [5]. Although these somatic mutations can be produced by accumulated unrepaired DNA errors linked to cell aging, UV exposure is in fact the main risk factor associated with CM development [6,7]. Additionally, other risk factors such as sex, individual’s skin phototype, age, and chronic or intermittent UV exposure contribute to increased CM risk [3,8].

In the last decades, NGS genomic studies have provided additional data regarding the genetic alterations that underline each CM subtype, and have highlighted CM as the cancer with the highest mutational burden and mutational heterogeneity [2,9]. On the whole, these studies show that somatic mutations in different driver genes (both oncogenes and tumor suppressor genes) are responsible for CM genesis, progression, or metastasis [1,10]. Additionally, recent works demonstrate that a subset of the mutations driving CM origin and development are already present in normal healthy skin as well as in nevi [4,11].

In this review, we present an overview of the genetic alterations found in normal skin and nevi, as well as those that lead to the development and progression of CM, focusing on UV exposure as the main CM risk factor. In addition, we also provide an overview of the current targeted therapies for the treatment of melanoma, including the challenge of acquired resistances.

## 2. Materials and Methods

An exhaustive literature search of publications in the English language available in PubMed and Scopus was performed from May to June 2022 using the following predefined keywords: “melanoma”, “cutaneous melanoma”, “nevus” “skin cancer”, “hereditary melanoma”, “melanoma progression”, “clonal evolution”, “sun-exposed normal skin”, “melanocytic nevi”, “targeted therapy”, “immunotherapy”, “acquired resistance”. Moreover, reference lists of relevant articles and available reviews were also examined to identify additional studies. Studies in languages other than English were excluded. No works were discarded a priori due to poor design or deficient quality of the data. The titles and abstracts of all studies were assessed first. If the titles and abstracts did not display enough information, the full text of the paper was also evaluated.

## 3. Genetic Alterations in Normal Skin and Nevi

### 3.1. Cancer-Driver Mutations in Normal Skin

Some CMs arise from visible precancerous lesions (nevi or dysplastic nevi) but others emerge in apparently “normal” skin zones. To date, several works have been conducted to study the molecular mechanisms that drive skin cancer onset from a set of somatic genetic mutations present in non-malignant cells [4,11,12].

Spontaneous somatic mutations are accumulated in cells throughout a person’s lifetime. These somatic mutations may be produced by unrepaired errors accumulated in the DNA after cell replication but also by exposure to mutagenic agents such as a number of specific chemical products as well as UVR [7]. Most of these somatic mutations do not have relevant effects, but some can lead to aging or cancer development [13]. In this context, age and skin phototype have a crucial role as main risk factors of skin cancer. Age has a strong impact on the buildup of somatic mutations in normal skin, and the rate of mutation accumulation is modulated as well by an individual’s skin phototype [4]. This reflects the incapacity of UV-sensitive individuals to keep their DNA from the damaging effects of UVR. Generally, individuals with high skin-phototypes (who tan easily) have a significantly lower number of somatic mutations accumulated in normal skin compared to people with low skin-phototypes (who usually cannot tan), the latter also having a higher susceptibility to developing skin cancers [4].

The burden of somatic UV-induced mutations in normal skin is surprisingly similar to that found in skin tumors [12]. UV-induced mutations cause a specific mutation pattern, known as UV mutational signature (signature 7), with mutations at dipyridine sites. C > T transitions represent more than 60% of the somatic mutational burden associated with UVR, while CC > TT more than 5% [14]. C > T and CC > TT mutations are more common on the non-transcribed strand of genes, but C > A transversions also occur [4,12]. Therefore, the most common mutations are single-nucleotide variants (SNVs) or dinucleotide variants (DNVs) with a quite low median variant allele fraction (VAF)—less than 5%—in normal healthy skin [11,12].

*TP53*-mutated clones of keratinocyte cells are regarded as one of the earliest signs of carcinogenesis induced by UVR [4]. However, mutations in the *TP53* gene are already present in normal skin. In addition, somatic mutations in genes such as *NOTCH1, NOTCH2, NOTCH3, FAT1* are also very frequent in normal sun-exposed skin [11,12]. NOTCH receptors are involved in the Notch signaling pathway, which regulates multiple processes related to cell–cell communication, cell fate specification, differentiation, proliferation, and survival, having been associated with carcinogenesis and tumor progression [15]. On the other hand, *FAT1* encodes a type of protocadherin, and it is probably the most commonly mutated cancer gene. *FAT1* regulates several signaling routes—e.g., Hippo, Wnt/β-catenin and, most importantly in melanoma, the MAPK/ERK pathway—involved in proliferation, migration, and invasion processes [16]. *TP53* codes for a tumor suppressor protein acting as a transcription factor regulating the expression of around 500 genes involved in cancer-relevant mechanisms such as DNA repair, apoptosis, cell cycle arrest, senescence, and metabolism. *TP53* mutations are universal across human cancers. Defects in *TP53* function lead to tumor development as well as anti-tumor drug resistance [17].

Recent studies have been carried out to characterize the genomic landscape of normal skin by comparing mutations between UV-exposed body sites and non-UV-exposed sites. In one of the studies, the characterization of somatic mutations in exposed and non-exposed normal-skin samples yielded a significantly higher number of clonal mutations in exposed skin (7.3 vs. 4.7 mutations respectively) for a set of selected genes [11]. Mutations in UV-exposed skin samples were especially enriched for *TP53* (*p* < 0.001), although genomic segments of *NOTCH1* were also significantly associated with UV exposure status. On the other hand, wide genomic regions were completely devoid of somatic mutations in non-exposed normal skin [11].

Mutations in *CDKN2A*, also a common melanoma and squamous cell carcinoma (SCC) driver gene, were not found to be under positive selection in normal human skin [12]. Maybe *CDKN2A* inactivation is more frequent in skin cancer clones and may confer a selective advantage in a more advanced cancer stage. Nevertheless, an additional study revealed a substantial excess of *CDKN2A* truncating mutations in normal skin samples from aged individuals [4]. Thus, more research in this field is needed.

Studies performed on individual healthy-skin melanocytes show that, as expected, melanocytes in exposed skin had numerous mutations related to UV-signatures (7a, 7b, 7c) as opposed to non-exposed melanocytes [18]. Interestingly, melanocytes from chronically-exposed skin had lower mutational burdens than melanocytes from intermittently-exposed skin [18]. This agrees with the fact that, unlike other skin cancer types, CM is comparatively more common in intermittently-exposed skin than in chronically-exposed skin. Furthermore, apparently healthy melanocytes flanking skin cancers seem to have similar mutational burdens to melanomas, but much higher mutational burdens than melanocytes from individuals lacking skin cancer [18]. Copy number alterations are comparatively unusual in melanocytes of normal skin. The genes more commonly mutated in individual melanocytes are inhibitors of the MAPK/ERK pathway: *NF1*, *RASA2,* and *CBL*, although gain-of-function alterations in *BRAF*, *NRAS*, and *MAP2K1* are also relatively common [18].

Above and beyond this, the effect of age on the accumulation of mutations also seems to be extremely relevant [4]. Somatic mutations in cancer-driving oncogenes such as *BRAF*, *PIK3CA*, *HRAS,* and *FGFR3* can be found in normal skin from elderly people, while younger people’s skins almost completely lack mutations in those genes. At the same time, UV-related mutational signatures are behind 70% of all mutations detected in subjects older than 63 years, but only represent around 47% of the mutational burden in younger people [4].

In brief, several studies have provided new insights into the genetic alterations that may cause the appearance of skin cancer from a set of somatic genetic mutations accumulated in skin cells [4,11,12]. However, at present, there are still fundamental gaps in our understanding of how skin cells progress to malignant cells and exactly which somatic mutations trigger the inception of a cancerous clone—and which do not.

### 3.2. Genetic Alterations of Benign and Dysplastic Nevus

#### 3.2.1. Benign Nevus

A benign nevus, melanocytic nevus, or mole, is a benign skin lesion caused by local proliferation of melanocytes. This can be present at birth (congenital melanocytic nevus) or appear later (acquired nevus) [19], both having unique genetic, histological, and clinical features.

Congenital nevi arise from melanocytes with proliferation-triggering somatic mutations produced in utero from the 5th to the 24th week of gestation. Consequently, their appearance is not mediated by UVR exposure [20]. These congenital nevi are categorized based on size into small, medium, and large or giant [20], showing a consistent relation between nevus size and mutation status [21]. The most commonly mutated gene in congenital nevi is *NRAS*, although *BRAF* mutations can also be present but at very low frequencies and only in small congenital nevi [21,22]. Usually there is no overlap regarding mutations in both *NRAS* and *BRAF*. Therefore, as seen in CM, simultaneous mutations in both of those two genes are extremely rare, both in nevi and in CM. Mutations in other genes frequently altered in CM (*TP53, CDKN2A, CDK4*) have not been detected in congenital nevi [21,22].

On the other hand, acquired nevi share both genetic alterations and UVR environmental influence with CM. Epidemiological and clinical studies have demonstrated that individuals with a fair phenotype (tendency to sunburn and poor tanning ability), who tend to show higher susceptibilities to CM, also present a higher predisposition to an increased number of acquired nevi [21].

Around 30–40% of CMs arise from a preexisting nevus, particularly in superficial spreading melanomas and in melanomas that develop in younger patients [23]. Therefore, it is not surprising that many of the genetic traits typical of CM are also found in acquired nevi.

Genetic analyses carried out in acquired nevi have revealed a low mutation burden in nevi compared to CM and have highlighted the recurrent mutations affecting codon 600 in *BRAF* (around 80%) or *NRAS* (20%) in this type of nevi [19,22,24]. Either *BRAF* or *NRAS* are usually mutated in the majority of acquired nevi, but hardly ever both at once. These mutations have been predicted to be clonal, suggesting a possible role as a founding genetic event [19,25]. *BRAF* V600 hotspot mutations are found in around 40–60% of all melanomas, and these mutations are present in about 67–90% of all acquired nevi [24,26]. Besides *BRAF* and *NRAS*, other genes found mutated in acquired nevi are *NOTCH2, PTPRD, PIK3C2G, SETD2,* and *ERBB4* [19]. However, there is a little overlap in these mutations among nevi, showing the great heterogeneity of the genetic profiles aside from *BRAF* and *NRAS* alterations.

In summary, seemingly *BRAF* and *NRAS* mutations represent common and early somatic events in benign nevi, but these changes alone are not sufficient to confer malignant behavior to melanocytes. Thus, other molecular events and signaling pathways ought to be involved in the genesis of CM.

#### 3.2.2. Dysplastic Nevus

A dysplastic nevus is a benign melanocytic proliferation typified by structural disarray and cytologic atypia, resulting from the interaction of genetic, environmental, and host factors. The occurrence of dysplastic nevi is a significant CM risk factor, and these types of nevi have been associated with practically 100% of familial and about 60% of sporadic cases of CM [27]. In addition, the risk of CM grows with increasing numbers of dysplastic nevi. While a single dysplastic nevus gives a twofold risk of melanoma, individuals with more than 10 have a twelvefold increased risk of CM [27]. UVR seems to be the chief environmental factor involved in the development of dysplastic nevi [27].

Several lines of evidence supported by genomic analyses suggest that dysplastic nevi occupy a sort of middle ground between benign nevi and CM, and reinforce the idea of these nevi as intermediate transitional stages in carcinogenesis [27]. The genetic alterations present in dysplastic nevi include: (i) intermediate microsatellite instability between benign nevi and CM; (ii) allelic losses at 1p, 9p, and 17p; (iii) DNA abnormalities similar to superficial spreading melanoma; (iv) V600E mutation in the *BRAF* gene; (v) loss of tumor suppressor genes (*TP53* and *CDKN2A)* and alterations in oncogenes (*RAS*), though at lower frequencies than in CM [21,27]. However, other studies analyzing dysplastic nevi have not found mutations in driver genes associated with melanoma, such as *CDKN2A, TP53, NF1, RAC1*, and *PTEN* [22,28]. Therefore, it seems that more research is needed to settle this issue.

Other works have provided more information regarding the similitudes and differences between dysplastic nevi and CMs concerning types of mutations and mutational signatures. On the one hand, dysplastic nevi show a significantly lower mutation rate than CMs, with an average of 18 and 34 mutations, respectively [28], suggesting that mutational burden leads towards a malignant state. C > T mutations are predominant (83%) both in dysplastic nevi and CM, while TC > TT changes were significantly more frequent in CMs (CC > CT or CC > TT changes did not show a difference). Since TC > TT and CC > CT are established UV signature mutations that are enriched in CM, the discrepancies found between samples could point towards different selective pressures acting on UV-induced mutations or on mechanisms of DNA-damage repair [28].

On the whole, these results show that, despite the presence of some genetic alterations, benign and dysplastic nevi harbor relatively stable genomes. Therefore, the progression to CM requires a higher mutational burden, mainly mutations affecting CM cancer-driving genes (oncogenes and tumor suppressor genes).

## 4. Main Cutaneous Melanoma Histologic Subtypes

CM arises from melanocytes in the epidermis that may develop either a vertical or a radial growth phase. Based on the characteristics of this growth phase, three main clinical subtypes of CM can be recognized: superficial spreading melanoma, lentigo maligna melanoma, and nodular melanoma.

### 4.1. Superficial Spreading Melanoma (SSM)

SSM is the most prevalent form of CM, accounting for around 41% of all melanomas [29]. It has a slow and radial growth pattern, so survival rates tend to be slightly higher than in other CMs, regardless of the stage at diagnosis (99.2% for local SSM) [30,31]. Often SSM originates from nevi, and it is clearly related to low cumulative sun damage [3]. Low-CSD CMs are mainly driven by *BRAF*-activating mutations, especially *BRAF* V600E [1], although the frequency of other *BRAF* mutations (V600K, K601E) or *NRAS* mutations increases with age [3]. SSM shows chromosomic alterations involving loss of 9p, 10q22.1, 10pter, 6q, 21q, and gains of 1q, 6p, 7, 8q, 17q, and 20q [32,33]. However, the mutation burden increases with tumor progression, with mutations always showing a strong relation with the UVR signature. Alterations in *TERT, CDKN2A, PTEN,* or *TP53* also play a part in the events that lead this process [1,3].

### 4.2. Lentigo Maligna Melanoma (LMM)

LMM is less frequent than SSM (2.7–14%) and it occurs mostly in elderly individuals [29,34]. LMM develops in heavily sun-exposed sites in 90% of cases, most commonly in the head or neck [34]. Therefore, this CM subtype is associated with high CSD. Like SSM, the prognosis of LMM patients is relatively good, with survival rates over 97% for early stages [35]. However, unlike SSM, LMM usually arises de novo rather than from nevi [31]. Regarding the LMM genomic landscape, it differs from non- or low-CSD melanomas and it is caused by a more miscellaneous set of MAPK/ERK pathway mutations [36]. LMM is characterized by a very high mutational load but with infrequent *BRAF* mutations [37]. Inactivating mutations in *NF1* (30%) are relatively common [1], as well as copy number increases of *CCND1* (20%), activating mutations of *KIT* (10%), and inactivating mutations of *TP53* and *ARID2* [3,10,37]. *TERT* promoter mutations are also fairly common [3,34]. This indicates that high- and low-CSD CMs are rather different molecular entities with distinct genetic profiles, progressing through different molecular pathways.

### 4.3. Nodular Melanoma (NM)

CM lesions that do not fulfill the features of either SSM or LMM are usually classified, almost by exclusion, as NM [3]. NM incidence has been shown to reach about 10–16% of all melanomas [29,38]. Unlike SSM and LMM, NM typically presents a vertical growth phase, and has relatively fast rates of growth (median 0.49 mm/month) compared with SSM or LMM (0.12 and 0.13 mm/month, respectively). Therefore, NM seems to be biologically more aggressive than the other CM histologic subtypes [31]. Consequently, the prognosis of patients is worse, with 5 year survival rates of 61.5% [39]. The genetic profile of NM overlaps with those of other melanoma subtypes. NM exhibits similar frequencies of *BRAF* mutations to SSM; however, it has been revealed to harbor significantly higher frequencies of *NRAS* mutations [3,10,31]. Conversely, other genes such as *TERT*, *ERBB3, NOTCH4*, *BCL2L12*, *SNX31*, *SSPO*, *ZNF560,* and *RPS6KA6* are significantly undermutated in NM in relation to SSM or LMM [31].

Table 1 displays the genetic alterations from early to metastatic CM for the three main subtypes of CM considered (SSM, LMM, and NM).

## 5. Cutaneous Melanoma Genetic Subtypes

In recent years, melanoma classifications no longer tend to follow the above histologic categories, but a molecular grouping characterized by the melanoma genetic profile. In that sense, it leans towards a classification regarding mutated driver genes instead of histological features.

Although hundreds of genes may happen to be mutated in CM, only some mutations are real “drivers” of the tumor, either as gain-of-function (activating) or loss-of-function (inactivating deleterious) mutations. The most important mutated genes contributing to melanoma development comprise *BRAF*, *NRAS*, and *NF1*, all of which may upregulate the MAPK/ERK pathway and promote cell proliferation [1,8]. Therefore, melanomas may be molecularly classified into four major genetic subtypes: *BRAF*-mutated, *RAS*-mutated, *NF1*-mutated and, if none of the three previous genes are mutated, triple-wildtype [40].

### 5.1. BRAF-Mutated Melanoma

*BRAF* is undoubtedly the most commonly mutated gene in CM [41], with its point mutations or genic fusions presenting in around 40% to 60% of cutaneous melanomas. Almost all histologic subtypes of melanoma harbor mutations in the *BRAF* gene, including superficial spreading melanoma, lentigo maligna melanoma, and other types of cutaneous and non-cutaneous melanoma (desmoplastic, acral, Spitz melanoma, etc.) [41]. The *BRAF* gene is an oncogene that, when mutated in melanoma, is always constitutively activated, and therefore upregulates the MAPK/ERK pathway. Only a few specific mutations allow for the *BRAF* gene to reach this constitutive activation, more than 99% of them on codon 600. The V600E change is easily the most frequent driver mutation in melanoma, especially in SSM, and it comprises up to 90% of all *BRAF* mutations in CMs, with V600K, V600R, and K601E making up most of the remaining 10% [41].

### 5.2. NRAS-Mutated Melanoma

*NRAS* is an oncogene that is mutated in around 15% to 25% of melanomas, and it is the most frequently mutated driver gene in CMs originated from congenital nevi, by a very long way [42]. As in *BRAF*, activating mutations in *NRAS* also stimulate the MAPK/ERK pathway, with more than 80% of *NRAS* mutations occurring at codon 61 [42]. Mutations in other genes of the same RAS family, such as *HRAS* or *KRAS*, are uncommon in CM, although they have also been identified in other melanoma subtypes (acral, mucosal, Spitz) [42].

### 5.3. NF1-Mutated Melanoma

*NF1*, a tumor suppressor gene, codes for neurofibromin, a protein that downregulates the MAPK/ERK pathway [42]. Therefore, unlike *BRAF* or *NRAS*, it is *NF1*’s loss of function which causes the activation of MAPK [42]. Inactivating mutations in the *NF1* gene are present in about 12–15% of melanomas, and are particularly frequent in high-CSD CMs, for example, lentigo maligna melanoma. This showcases the high burden of the UV mutational signature in *NF1*-mutant CMs [42]. Though *NF1* mutations ordinarily happen in CMs lacking mutations in *BRAF* or *NRAS*, nearly 4% of melanomas with mutations in *BRAF* or *NRAS* also exhibit *NF1* mutations [42].

### 5.4. Triple-Wildtype Melanomas

Between 25% and 35% of all melanomas lack mutations in *BRAF*, *NRAS,* and *NF1.* These melanomas are classified as triple-negative o triple-wildtype melanomas. In addition, these triple-negative tumors normally belong to the non-CSD category (melanomas on skin with little or no chronic sun-induced damage) [40].

Though both *BRAF* and *NRAS* mutations are rather pervasive and may be present in many CM types, on the whole, associations between the mutated driver gene and CSD type can be established. In general, low-CSD CMs have a mild mutational burden, are more common in intermittently-exposed skin areas, and often harbor *BRAF* V600E mutations [40]. On the other hand, high-CSD CMs show a very high mutational burden, are more prevalent in chronically-exposed body sites, and are linked to *NRAS* mutations as well as *BRAF* non-V600E mutations. Finally, as stated above, *NF1* mutations are associated with high-CSD CMs, while the triple-negative type is usually present in very low or non-CSD melanomas [40].

As already mentioned, *BRAF* and *RAS* mutations are usually mutually exclusive, although they can be found together rarely in different CM cell populations [43]. Additionally, other singular genomic alterations can be linked to each one of the stages related to the origin, progression, and metastasis of the tumor [43]. Figure 1 shows the genetic alterations that characterize each evolutionary stage of CM. These include normal skin, benign and dysplastic nevi, and early, advanced, and metastatic CM.

## 6. Genetic Evolution of Cutaneous Melanoma

### 6.1. Melanoma In Situ or Stage 0 (Early Tumorigenic Mutations)

Melanoma in situ represents the first stage of CM carcinogenesis, in which cancerous cells are found only in the epidermis (the upper layer of the skin). There are a handful of early genetic events linked to the development of melanoma in situ, including mutations in *BRAF*, *NRAS,* or *NF1*. Hence, early events on these genes (especially *BRAF* and *NRAS*) come across as the real initiators of CM, since parallel mutation rates of these two genes are found both in benign precancerous lesions and in early stages of CM [10]. Loss of function in *NF1*, leading to the activation of MAPK/ERK signaling, also upregulates PI3K/AKT. The activation of the PI3K/AKT pathway promotes proliferation, metabolism, angiogenesis, and survival, as well as mTOR oncogenic signaling [42]. *NF1* loss often happens in melanomas lacking mutations in *BRAF* or *NRAS*, although in 4% of CM these three mutations can coexist [42,44]. On the other hand, *KIT* expression is upregulated in early CM but downregulated in later stages of the disease [1,45]. These data put forward the important early role of *KIT* (another oncogene involved in melanoma, though less frequently mutated in CM) in oncogenesis, and its silencing by other driver genes in advanced melanoma stages (Figure 1).

### 6.2. Advanced and Metastatic CM

The evolution of CM from superficial (radial growth phase) to invasive (vertical growth phase) is mainly associated with activating *TP53* mutations, *PTEN* loss, and *CDKN2A* inactivation, which trigger apoptotic inhibition and cell proliferation. *CDKN2A* inactivation is present in over 90% of melanomas, especially in advanced stages. Conversely, *KIT* expression is markedly downregulated in later stages of the disease (unlike in the early stages) [1]. Lastly, *TERT* mutations have been found in 30% to 70% of sporadic late-stage melanomas, especially NM and SSM [46,47].

After local and regional invasion, metastasis can be produced by two parallel processes, one that includes sequential stages: primary tumor–regional lymph nodes–blood vessels–distant organs; and another involving circulating tumor cells. This second process could explain why distant metastases are usually larger and seem to appear earlier [1]. Although the genomic evolution of metastasis in CM is not as clear as in other melanoma subtypes, several studies reinforce the idea that mutations in *EGFR4* and *NMDAR2*, amplifications of *MITF* and *MET*, and loss of *PTEN* can be regarded as metastasis drivers [10]. Biallelic loss of *CDKN2A* also promotes metastasis by increasing the transcription factor BRN2 [48]. The role of *TERT* in CM metastasis has also been studied. *TERT* promoter mutations occur more commonly in aggressive CM forms and tend to be associated with distant metastases and patient mortality. In addition, the concurrence of *BRAF* mutations with *TERT* promoter mutations in CM is linked to a higher tumor aggressiveness [49].

On the other hand, invasion and spread of melanoma (metastasis) are also associated with mutations in genes involved in cell adhesion mechanisms (cadherins, integrins) that are in charge of cell migration, tissue organization, and organogenesis [50]. Thus, their dysregulation contributes to cancer cell invasion and migration. In melanoma, the progress from the radial growth phase to the vertical growth phase is denoted by R-cadherin loss together with N-cadherin gain [51,52]. Normal melanocytes are monitored by keratinocytes via R-cadherin, so mutations in these genes let melanocytes free from the regulation of keratinocytes and therefore may interact with stromal cells, contributing to melanoma cancer cell invasion [51,52].

Finally, melanoma genetic evolution towards metastasis is led by whole genome doubling (WGD) and large-scale aneuploidy with extensive loss of heterozygosity (LOH) in late distant metastasis (>50% compared to earlier disease) [43]. Therefore, loss of genomic integrity is a crucial factor in the selection of advantageous cancerous cell clones during melanoma evolution [53]. Copy number alterations (CNAs), including deletions and amplifications, appear rarely in nevus and early melanoma stages, but are widespread in invasive and metastatic melanoma [43,54]. While early mutations are obviously driven mostly by UV mutational signatures (signatures 7a, 7b and 7c), mutations in late-stage melanoma typically possess non-UV signatures. Unsurprisingly, this corroborates the important role of UV-induced mutations in CM early stages, as well as the prominence of non-UV signature events in melanoma progression and metastasis [55].

## 7. New Melanoma Therapies in Present-Day Clinical Practice

Until the last decade, the treatment options for advanced or metastatic melanoma included surgical resection, systemic chemotherapy (dacarbazine), and elevated doses of interleukin 2 (IL-2). Despite these treatments, metastatic disease had a poor prognosis, with very low survival rates (less than 12 months) [55,56]. Nevertheless, in recent years, the emergence of NGS has provided unprecedented insights regarding the somatic genetic alterations of melanoma and the molecular mechanisms leading to its origin and progression [57]. This fact has meant a revolution in melanoma management through the development of novel targeted therapies and the advancement of immunotherapy, resulting in important effects on melanoma survival rates.

### 7.1. Targeted Therapies

The arrival of NGS has meant a major breakthrough in personalized medicine, since it has facilitated the uncovering of many cancer driver mutations, resistance mechanisms to treatments, determination of mutational burden and germline mutations, building the basis of a new strategy in cancer management [58]. In the context of melanoma, prevalent mutations in *BRAF, NRAS, MEK,* and *KIT* oncogenes have already made possible the development of specific targeted therapies directed at the MAPK/ERK pathway, often constitutionally activated in melanoma [59]. Table 2 shows the different targeted-therapy drugs that have been developed in the last decade for melanoma, their associated molecular targets and their clinical status (FDA approved vs. clinical trial).

Although targeted therapies for advanced and metastatic melanoma have shown evident advances in the response and survival rates compared to conventional treatments, differences in clinical outcomes can be observed depending on the targeted therapies administered [60].

The targeted therapies currently being used, already approved by the FDA, concentrate on mutations in two proteins, B-Raf and MEK, both part of the MAPK/ERK pathway [60]. In addition, other therapies in different stages of development (clinical trial or preclinical studies) also focus on other components of the Ras family (N-Ras H-Ras, and K-Ras), Met, VEGFR, PI3K, and especially Kit [60]. Mutations in these proteins cause the stimulation of three different pathways, MAPK/ERK, PI3K/AKT, and JAK/STAT, which have a profound effect on cell proliferation and cell survival. Table 2 summarizes the currently approved targeted treatments for advanced melanoma, as well as the drugs that at this time are in clinical trials.

Vemurafenib was the first inhibitor specifically blocking B-Raf that was approved by the FDA. Two other B-Raf inhibitors, dabrafenib and encorafenib, were subsequently developed, and currently all three are being used to treat advanced *BRAF*-mutated melanomas. Of all approved B-Raf inhibitors, encorafenib presents a more intense inhibition of B-Raf, blocking the MAPK/ERK signaling pathway more efficiently and thus showing a stronger anti-cancer action [60].

Next, three MEK inhibitors, cobimetinib, trametinib, and binimetinib, were also approved for clinical use in advanced *BRAF*-positive melanoma. MEK is another member of the MAPK/ERK signaling pathway commonly activated in melanoma, and is a kinase protein located downstream of B-Raf [60].

However, the most efficient targeted therapies have proved to be a combination of both a B-Raf and a MEK inhibitor [61]. The three B-Raf/MEK combinations available (vemurafenib with cobimetinib, dabrafenib together with trametinib, and finally encorafenib with binimetinib) have greatly bettered the clinical effectiveness of *BRAF* V600-mutant advanced melanomas compared to B-Raf monotherapy, with improved progression-free survival and longer overall survival rates [62,63].

In conclusion, the analyses derived from different clinical studies indicate that combined B-Raf inhibition plus MEK inhibition improves efficacy compared to B-Raf inhibition alone for the treatment of advanced/metastatic melanoma harboring the *BRAF* V600 mutation, with three different combinations approved by the FDA (see above). Finally, other drugs classified as MEK inhibitors (selumetinib or pimasertinib) [64,65], Kit inhibitors (imatinib, sunitinib, dasatinib, and nilotinib) [66] or even Met or VEGFR inhibitors are being tested in various clinical trials but have not yet been approved by the FDA. Consequently, at present, B-Raf and MEK inhibitor combined treatment is the preferable choice for patients with *BRAF* V600-mutant melanoma [67].

**Table 2 life-12-01339-t002:** Current targeted therapies for metastatic melanoma treatment.

Targets (Biomarkers)	Drug (Inhibitor)	Reference
**FDA approved**		
B-Raf V600	Vemurafenib	[68,69,70,71]
	Dabrafenib	[60,61,72]
	Encorafenib	[62,73]
MEK	Trametinib	[72,74]
	Binimetinib	[62,73]
	Cobimetinib	[71,75]
**Clinical trials**		
MEK	Selumetinib	[64]
	Pimasertib	[65]
Kit	Imatinib	[66]
	Sunitinib	[66]
	Dasatinib	[66]
	Nilotinib	[66]
Met	Tivantinib	[76,77]
	Cabozantinib	[65,78]
VEGFR	Axitinib	[79,80]
	Sorafenib	[81]

Note: B-Raf: B-Raf serine/threonine kinase; MEK: mitogen-activated protein kinase; Kit: receptor-tyrosine kinase Kit.

However, these treatments are not curative, since most patients end up developing drug resistance and, subsequently, melanoma relapse. Drug resistance to B-Raf inhibitors was partially palliated and delayed when the B-Raf/MEK combined therapy was introduced, but the occurrence of resistance to drug therapy is still the main limitation of B-Raf monotherapy or combined B-Raf/MEK inhibitor therapy for advanced and metastatic melanoma [61].

### 7.2. Mechanisms of Drug Resistance in Melanoma

Melanoma cells use different molecular routes to generate resistance to B-Raf targeted therapies. These mechanisms end up reactivating the MAPK/ERK pathway even when B-Raf signaling is inhibited. At the moment, several MAPK/ERK reactivation routes have been identified, including the following: upregulation of *PDGFRB*; upregulation of *IGF1R*; gain of function at the *PI3K* level or COT kinase; *MEK1* mutations; presence of RAS-independent *BRAF* mutations (mainly *NRAS*); altered *BRAF* amplification or splice- variant *BRAF* expression; amplification of *MITF*; and loss of function of *NF1* [59]. Out of all these, the most common are overexpression of *PI3K*, *NRAS* mutations, amplification of *MITF*, the appearance of *MEK1* mutations, and mutated *BRAF* amplification signaling [59].

The selection of new *NRAS* mutations is the principal cause of melanoma reactivation in *BRAF*- or *NRAS*-mutant cases, explaining why *BRAF*-mutant or *NRAS*-mutant melanomas exhibit the largest resistance to B-Raf inhibitors [1]. Also, several studies have shown that *BRAF* amplification signaling in *BRAF* V600E-mutated melanomas can cause spontaneous dimerization of the ensuing molecule, and this dimerization is able to reactivate the MAPK/ERK pathway [59]. Moreover, augmented *BRAF* copy numbers have also been detected in around 20% of cases of B-Raf inhibitor therapy resistance in melanoma [59]. *MEK1* mutations E203 and Q56 have been identified in melanoma tissue samples of patients that developed resistance to vemurafenib [82]. On the other hand, MITF may cause resistance to B-Raf/ERK inhibitors via several routes, including increased survival signaling and alterations of metabolism [83]. Upregulated *MITF* expression is related to inherent B-Raf inhibitor resistance, and *MITF* amplification is associated with *BRAF*-positive melanomas [83].

Regarding MEK inhibitors, there are also frequent limitations due to several mechanisms of resistance. The most obvious is amplification of B-Raf, located upstream of MEK, which causes MEK hyperactivation and ends up lessening the efficacy of MEK inhibitors [84]. B-Raf amplification has also shown the ability to activate K-Ras, making melanoma cells less sensitive to MEK inhibitors [85]. Furthermore, some MEK1 mutations—in codon P124 or codon Q56—grant resistance to treatment with both B-Raf and MEK blocking agents [82]. In addition, upregulation of the STAT3 pathway has been linked to acquired drug resistance to MEK inhibitors via loss of BIM, a component of the Bcl-2 family usually involved in tumor suppression [86]. Finally, amplified expression of ITF-2, coding for a transcription factor implicated in lymphocyte growth and maturation, has also been involved in the mechanisms leading to resistance to MEK inhibitors [87]. Transcription of ITF-2 is directed by the Wnt pathway and, predictably, the expression of ITF-2 was found to be considerably activated in melanoma cell lines that presented resistance to MEK inhibitors [87]. In fact, Wnt signaling upregulation through ERK has also been detected in MEK-resistant melanomas [87].

### 7.3. Immunotherapy

In the last few years, the discovery of the role of immune checkpoint molecules, e.g., cytotoxic T-lymphocyte-associated antigen 4 (CTLA-4), programmed cell-death protein 1 (PD1), and PD-ligand 1 (PD-L1), has enabled the detection of immune checkpoint inhibitors with promising results for melanoma treatment [67]. Several immunotherapies such as CTLA-4 inhibitors (ipilimumab), PD-1 inhibitors (nivolumab, dostarlimab, and pembrolizumab), and PD-L1 inhibitors (atezolizumab) have been approved by the FDA.

When compared to traditional cytokine-based treatment, these blockers have shown better patient tolerance and have managed to lengthen overall survival thanks to immunological memory [67,88]. In spite of the efficiency of these treatments, just a few melanoma patients have attained long-lasting clinical responses with monotherapy, so the combination with other immune checkpoint inhibitors, B-Raf/MEK inhibitors, or other treatments (e.g., chemotherapy, radiotherapy) has become the best option for the treatment of melanoma [89,90]. Several studies using combined therapies have showed that dual treatments provide greater benefit with higher overall and progression-free survival rates. For instance, the clinical trial (phase III) IMspire150, designed to assess combination therapies with BRAF and MEK inhibitors (vemurafenib, cobimetinib) and immune checkpoint therapy (atezolizumab) in BRAF V600-positive advanced or metastatic melanoma (*n* = 514), showed the value of the combined treatment. Moreover, the addition of atezolizumab was safe and tolerable and considerably reduced melanoma progression [91]. Other studies conducted with combined therapy, using ipilimumab and nivolumab, also provided higher overall survival rates at 3 years, and progression-free survival of 11.5 months, compared to 6.9 months of the nivolumab only group, and 2.9 months of the ipilimumab monotherapy group.

On the downside, combined therapy produces greater toxicity than previous treatments: 59% of patients treated with dual immunotherapy had immune-related adverse complications, compared with those treated only with ipilimumab (28%) or nivolumab (21%) [92,93].

It seems clear that combined immunotherapies are more efficient than monotherapies, and that these treatments have provided patients and healthcare professionals with a new powerful tool to deal with late-stage melanoma. However, long-term successful treatment is still relatively infrequent for patients with advanced melanoma, since many of them end up developing acquired resistance and subsequently relapse [94].

In summary, the role of targeted therapies as an alternative to traditional treatments and the advent of immunotherapy have completely transformed and updated the treatment of advanced and metastatic melanoma. Although these new treatments, especially combined therapies, are more successful and less toxic compared to the traditional ones, the new targeted therapies are actually not particularly efficient in the long run, due to the appearance of drug resistance and the ensuing relapse. Therefore, in order to improve the prognosis in advanced and metastatic melanoma patients, more research and new management strategies will be required for the development of better therapies.

## 8. Conclusions and Future Perspectives

The application of NGS in the oncology field, specifically in the study of melanoma, has revealed the main genetic alterations linked to the origin, development, and progression of this disease. As a result, this information has promoted the development of more effective and safer treatments for advanced melanomas, fueling the appearance of new targeted therapies as well as successful immunotherapy treatments.

However, there are still knowledge gaps related to clonal selection and the mechanisms underlying malignant transformation from normal cells or nevi to CM, as well as in the molecular mechanisms that lead to acquired targeted-therapy resistance. The high heterogeneity present in CM, and the complex relationships established between cancer metabolism, tumor microenvironment, and the immune system have made it difficult to fully characterize all the evolutionary stages of this tumor from its onset to metastasis.

Consequently, more studies are required to investigate these events from a comprehensive approach, in order to provide new information that improves the diagnosis, follow-up, or treatment of CM through the identification of new target molecules for drug discovery. For instance, the implementation of validated genetic biomarkers in the clinical setting—e.g., as a screening method for early diagnosis—would make it possible to treat CM in the early stages, reducing the high mortality rates associated with this tumor.

On the other hand, it is worth highlighting the role of UVR as the main risk factor in CM pathogenesis, evidenced by the omnipresence of the UV mutational signature in CM tumor tissue (e.g., 7a, 7b, 7c). For this reason, preventive measures related to sun protection are crucial to avoid CM development, especially in sun-sensitive individuals.

## Figures and Tables

**Figure 1 life-12-01339-f001:**
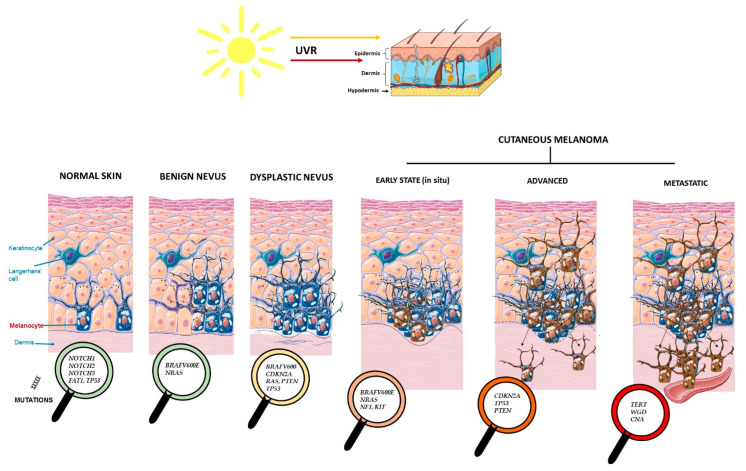
Driver genes involved in CM development and progression from normal healthy skin and nevus.

**Table 1 life-12-01339-t001:** Genomic mutations throughout the progression of cutaneous melanoma. The color scheme indicates: activating mutations in oncogenes, loss-of-function mutations in tumor suppressor genes.

	Cutaneous Melanoma		
	SSM	LMM	NM
**Cumulative Sun Damage**	Low-CSD melanomas	High-CSD melanomas	High- and Low-CSD
**Initial stages**	*BRAF, NRAS* Less frequent: *MAP2K1, CTNNB1, PRKCA** APC, BAP1, PRKAR1A*	* NRAS, BRAF, KIT, * * NF1 *	*BRAF, NRAS, **NF1* Less frequent: *ERBB3, NOTCH4, BCL2L12*
**Malignant transformation**	* TERT, * * CDKN2A, TP53, PTEN *	* RAC1, TERT, * * CDKN2A, TP53, PTEN, ARID2 *	* TERT, * * CDKN2A, TP53 *
**Metastatic phase**	Duplications of entire genome, CNVs and aneuploidy	Duplications of entire genome, CNVs and aneuploidy	Genome duplications, CNVs and aneuploidy

Abbreviations: CNV: Copy number variation; LMM: lentigo maligna melanoma; NM: nodular melanoma; SSM: superficial spreading melanoma.

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
