# Peer review of "UV-Induced Somatic Mutations Driving Clonal Evolution in Healthy Skin, Nevus, and Cutaneous Melanoma"

_life, 2022, doi:10.3390/life12091339_

Round 1

Reviewer 1 Report

The manuscript entitled:" UV-induced somatic mutations driving clonal evolution in 3 healthy skin, nevus and cutaneous melanoma" focused on a systemic revision of literature data about the clonogenic evaluation of molecular landscape in melanoma patients requires several major integrations to be accepted for the publication:

- In the introduction section, the authors summarize the molecular features of melanoma patients. As regards, technical and clinical aspects related to melanoma patients should be exaustively investigated.

- In the methodological section, please, could the authors also show the results of thier literature data search able to identify clinically impacting manuscript?

- In the discussion section, few details are reported about the description of molecular features of each clinical stage. In my opinion, the authors should investigate in details the data about this points. In addition, i would also encouraged the authors to analyze other molecular events that drastically impact on melanoma patients (PD-L1, TMB). How this molecular signature may be infleunced by NGS approach and could play a pivotal role in the decision making process for melanoma patients

- Generally, clinical section is too poor in relation to emerging molecular biomarkers that should be detected in evolving melanoma disease. Accordingly,m i would strongly suggest to inevstigate in a dedictaed pragraph the emerging molecular biomarkers and the clinical relationship in this patients setting.

Author Response

Response to the referee’s comments

Reviewer 1

According to the comments provided by Reviewer 1, several changes have been performed in the manuscript entitled "UV-induced somatic mutations driving clonal evolution in healthy skin, nevus and cutaneous melanoma” All these changes have been written in red, so there are easy to locate in the manuscript.

Below we respond point-by-point to the comments of the reviewer:

Comment 1:  In the introduction section, the authors summarize the molecular features of melanoma patients. As regards, technical and clinical aspects related to melanoma patients should be exaustively investigated.

The clinical aspects related to the management of melanoma patients have been included in a whole new section (section 7) dedicated to more clinical issues, mainly new advanced treatments and development of acquired resistances to those therapies. This new section is entitled: ”7. New melanoma therapies in the present-day clinical practice”. We believe that this new section covers all aspects mentioned in the reviewer’s Comment 1.

Comment 2: In the methodological section, please, could the authors also show the results of their literature data search able to identify clinically impacting manuscript?

The initial manuscript includes key references on basic and translational studies performed primarily on human samples focused on the characterization of the genetic mutations that lead to the development and progression of melanoma. Articles that include data on clinical trials and clinical studies have also been included as new references in the new section 7.

Newly added clinical references (after inclusion of section 7) are the following:

  1. Morganti, S.; Tarantino, P.; Ferraro, E.; D’Amico, P.; Duso, B.A.; Curigliano, G. Next Generation Sequencing (NGS): A Revolutionary Technology in Pharmacogenomics and Personalized Medicine in Cancer. In Translational Research and Onco-Omics Applications in the Era of Cancer Personal Genomics; Ruiz-Garcia, E., Astudillo-de la Vega, H., Eds.; Advances in Experimental Medicine and Biology; Springer International Publishing: Cham, 2019; pp. 9–30 ISBN 978-3-030-24100-1.
  2. Patel, M.; Eckburg, A.; Gantiwala, S.; Hart, Z.; Dein, J.; Lam, K.; Puri, N. Resistance to Molecularly Targeted Therapies in Melanoma. Cancers (Basel) 2021, 13, 1115, doi:10.3390/cancers13051115.
  3. Ryu, S.; Youn, C.; Moon, A.R.; Howland, A.; Armstrong, C.A.; Song, P.I. Therapeutic Inhibitors against Mutated BRAF and MEK for the Treatment of Metastatic Melanoma. Chonnam Med J 2017, 53, 173–177, doi:10.4068/cmj.2017.53.3.173.
  4. Robert, C.; Grob, J.J.; Stroyakovskiy, D.; Karaszewska, B.; Hauschild, A.; Levchenko, E.; Chiarion Sileni, V.; Schachter, J.; Garbe, C.; Bondarenko, I.; et al. Five-Year Outcomes with Dabrafenib plus Trametinib in Metastatic Melanoma. N Engl J Med 2019, 381, 626–636, doi:10.1056/NEJMoa1904059.
  5. Dummer, R.; Ascierto, P.A.; Gogas, H.J.; Arance, A.; Mandala, M.; Liszkay, G.; Garbe, C.; Schadendorf, D.; Krajsova, I.; Gutzmer, R.; et al. Encorafenib plus Binimetinib versus Vemurafenib or Encorafenib in Patients with BRAF-Mutant Melanoma (COLUMBUS): A Multicentre, Open-Label, Randomised Phase 3 Trial. Lancet Oncol 2018, 19, 603–615, doi:10.1016/S1470-2045(18)30142-6.
  6. Ascierto, P.A.; Dréno, B.; Larkin, J.; Ribas, A.; Liszkay, G.; Maio, M.; Mandalà, M.; Demidov, L.; Stroyakovskiy, D.; Thomas, L.; et al. 5-Year Outcomes with Cobimetinib plus Vemurafenib in BRAFV600 Mutation-Positive Advanced Melanoma: Extended Follow-up of the CoBRIM Study. Clin Cancer Res 2021, 27, 5225–5235, doi:10.1158/1078-0432.CCR-21-0809.
  7. Robert, C.; Dummer, R.; Gutzmer, R.; Lorigan, P.; Kim, K.B.; Nyakas, M.; Arance, A.; Liszkay, G.; Schadendorf, D.; Cantarini, M.; et al. Selumetinib plus Dacarbazine versus Placebo plus Dacarbazine as First-Line Treatment for BRAF-Mutant Metastatic Melanoma: A Phase 2 Double-Blind Randomised Study. Lancet Oncol 2013, 14, 733–740, doi:10.1016/S1470-2045(13)70237-7.
  8. Lebbé, C.; Dutriaux, C.; Lesimple, T.; Kruit, W.; Kerger, J.; Thomas, L.; Guillot, B.; Braud, F. de; Garbe, C.; Grob, J.-J.; et al. Pimasertib Versus Dacarbazine in Patients With Unresectable NRAS-Mutated Cutaneous Melanoma: Phase II, Randomized, Controlled Trial with Crossover. Cancers (Basel) 2020, 12, E1727, doi:10.3390/cancers12071727.
  9. Pham, D.D.M.; Guhan, S.; Tsao, H. KIT and Melanoma: Biological Insights and Clinical Implications. Yonsei Med J 2020, 61, 562–571, doi:10.3349/ymj.2020.61.7.562.
  10. Ralli, M.; Botticelli, A.; Visconti, I.C.; Angeletti, D.; Fiore, M.; Marchetti, P.; Lambiase, A.; de Vincentiis, M.; Greco, A. Immunotherapy in the Treatment of Metastatic Melanoma: Current Knowledge and Future Directions. J Immunol Res 2020, 2020, 9235638, doi:10.1155/2020/9235638.
  11. Kim, G.; McKee, A.E.; Ning, Y.-M.; Hazarika, M.; Theoret, M.; Johnson, J.R.; Xu, Q.C.; Tang, S.; Sridhara, R.; Jiang, X.; et al. FDA Approval Summary: Vemurafenib for Treatment of Unresectable or Metastatic Melanoma with the BRAFV600E Mutation. Clin Cancer Res 2014, 20, 4994–5000, doi:10.1158/1078-0432.CCR-14-0776.
  12. Chapman, P.B.; Hauschild, A.; Robert, C.; Haanen, J.B.; Ascierto, P.; Larkin, J.; Dummer, R.; Garbe, C.; Testori, A.; Maio, M.; et al. Improved Survival with Vemurafenib in Melanoma with BRAF V600E Mutation. New England Journal of Medicine 2011, 364, 2507–2516, doi:10.1056/NEJMoa1103782.
  13. Kim, A.; Cohen, M.S. The Discovery of Vemurafenib for the Treatment of BRAF-Mutated Metastatic Melanoma. Expert Opin Drug Discov 2016, 11, 907–916, doi:10.1080/17460441.2016.1201057.
  14. Álamo, M. del C.; Ochenduszko, S.; Crespo, G.; Corral, M.; Oramas, J.; Sancho, P.; Medina, J.; Garicano, F.; López, P.; Balea, B.C.; et al. Durable Response to Vemurafenib and Cobimetinib for the Treatment of BRAF-Mutated Metastatic Melanoma in Routine Clinical Practice. OTT 2021, 14, 5345–5352, doi:10.2147/OTT.S325208.
  15. Wahid, M.; Jawed, A.; Mandal, R.K.; Dar, S.A.; Akhter, N.; Somvanshi, P.; Khan, F.; Lohani, M.; Areeshi, M.Y.; Haque, S. Recent Developments and Obstacles in the Treatment of Melanoma with BRAF and MEK Inhibitors. Critical Reviews in Oncology/Hematology 2018, 125, 84–88, doi:10.1016/j.critrevonc.2018.03.005.
  16. Rose, A.A.N. Encorafenib and Binimetinib for the Treatment of BRAF V600E/K-Mutated Melanoma. Drugs Today (Barc) 2019, 55, 247–264, doi:10.1358/dot.2019.55.4.2958476.
  17. Hoffner, B.; Benchich, K. Trametinib: A Targeted Therapy in Metastatic Melanoma. J Adv Pract Oncol 2018, 9, 741–745.
  18. Boespflug, A.; Thomas, L. Cobimetinib and Vemurafenib for the Treatment of Melanoma. Expert Opin Pharmacother 2016, 17, 1005–1011, doi:10.1517/14656566.2016.1168806.
  19. Rimassa, L.; Assenat, E.; Peck-Radosavljevic, M.; Pracht, M.; Zagonel, V.; Mathurin, P.; Rota Caremoli, E.; Porta, C.; Daniele, B.; Bolondi, L.; et al. Tivantinib for Second-Line Treatment of MET-High, Advanced Hepatocellular Carcinoma (METIV-HCC): A Final Analysis of a Phase 3, Randomised, Placebo-Controlled Study. Lancet Oncol 2018, 19, 682–693, doi:10.1016/S1470-2045(18)30146-3.
  20. Kumar, S.R.; Gajagowni, S.; Bryan, J.N.; Bodenhausen, H.M. Molecular Targets for Tivantinib (ARQ 197) and Vasculogenic Mimicry in Human Melanoma Cells. European Journal of Pharmacology 2019, 853, 316–324, doi:10.1016/j.ejphar.2019.04.010.
  21. Daud, A.; Kluger, H.M.; Kurzrock, R.; Schimmoller, F.; Weitzman, A.L.; Samuel, T.A.; Moussa, A.H.; Gordon, M.S.; Shapiro, G.I. Phase II Randomised Discontinuation Trial of the MET/VEGF Receptor Inhibitor Cabozantinib in Metastatic Melanoma. Br J Cancer 2017, 116, 432–440, doi:10.1038/bjc.2016.419.
  22. Zhang, X.; Fang, X.; Gao, Z.; Chen, W.; Tao, F.; Cai, P.; Yuan, H.; Shu, Y.; Xu, Q.; Sun, Y.; et al. Axitinib, a Selective Inhibitor of Vascular Endothelial Growth Factor Receptor, Exerts an Anticancer Effect in Melanoma through Promoting Antitumor Immunity. Anticancer Drugs 2014, 25, 204–211, doi:10.1097/CAD.0000000000000033.
  23. Sheng, X.; Yan, X.; Chi, Z.; Si, L.; Cui, C.; Tang, B.; Li, S.; Mao, L.; Lian, B.; Wang, X.; et al. Axitinib in Combination With Toripalimab, a Humanized Immunoglobulin G4 Monoclonal Antibody Against Programmed Cell Death-1, in Patients With Metastatic Mucosal Melanoma: An Open-Label Phase IB Trial. J Clin Oncol 2019, 37, 2987–2999, doi:10.1200/JCO.19.00210.
  24. Liu, L.; Cao, Y.; Chen, C.; Zhang, X.; McNabola, A.; Wilkie, D.; Wilhelm, S.; Lynch, M.; Carter, C. Sorafenib Blocks the RAF/MEK/ERK Pathway, Inhibits Tumor Angiogenesis, and Induces Tumor Cell Apoptosis in Hepatocellular Carcinoma Model PLC/PRF/5. Cancer Res 2006, 66, 11851–11858, doi:10.1158/0008-5472.CAN-06-1377.
  25. Emery, C.M.; Vijayendran, K.G.; Zipser, M.C.; Sawyer, A.M.; Niu, L.; Kim, J.J.; Hatton, C.; Chopra, R.; Oberholzer, P.A.; Karpova, M.B.; et al. MEK1 Mutations Confer Resistance to MEK and B-RAF Inhibition. Proc Natl Acad Sci U S A 2009, 106, 20411–20416, doi:10.1073/pnas.0905833106.
  26. Smith, M.P.; Brunton, H.; Rowling, E.J.; Ferguson, J.; Arozarena, I.; Miskolczi, Z.; Lee, J.L.; Girotti, M.R.; Marais, R.; Levesque, M.P.; et al. Inhibiting Drivers of Non-Mutational Drug Tolerance Is a Salvage Strategy for Targeted Melanoma Therapy. Cancer Cell 2016, 29, 270–284, doi:10.1016/j.ccell.2016.02.003.
  27. Corcoran, R.B.; Dias-Santagata, D.; Bergethon, K.; Iafrate, A.J.; Settleman, J.; Engelman, J.A. BRAF Gene Amplification Can Promote Acquired Resistance to MEK Inhibitors in Cancer Cells Harboring the BRAF V600E Mutation. Sci Signal 2010, 3, ra84, doi:10.1126/scisignal.2001148.
  28. Wang, Y.; Van Becelaere, K.; Jiang, P.; Przybranowski, S.; Omer, C.; Sebolt-Leopold, J. A Role for K-Ras in Conferring Resistance to the MEK Inhibitor, CI-1040. Neoplasia 2005, 7, 336–347, doi:10.1593/neo.04532.
  29. Dai, B.; Meng, J.; Peyton, M.; Girard, L.; Bornmann, W.G.; Ji, L.; Minna, J.D.; Fang, B.; Roth, J.A. STAT3 Mediates Resistance to MEK Inhibitor through MicroRNA MiR-17. Cancer Res 2011, 71, 3658–3668, doi:10.1158/0008-5472.CAN-10-3647.
  30. Hur, E.-H.; Goo, B.-K.; Moon, J.; Choi, Y.; Hwang, J.J.; Kim, C.-S.; Bae, K.S.; Choi, J.; Cho, S.Y.; Yang, S.-H.; et al. Induction of Immunoglobulin Transcription Factor 2 and Resistance to MEK Inhibitor in Melanoma Cells. Oncotarget 2017, 8, 41387–41400, doi:10.18632/oncotarget.17866.
  31. Brahmer, J.R.; Tykodi, S.S.; Chow, L.Q.M.; Hwu, W.-J.; Topalian, S.L.; Hwu, P.; Drake, C.G.; Camacho, L.H.; Kauh, J.; Odunsi, K.; et al. Safety and Activity of Anti-PD-L1 Antibody in Patients with Advanced Cancer. N Engl J Med 2012, 366, 2455–2465, doi:10.1056/NEJMoa1200694.
  32. Gotwals, P.; Cameron, S.; Cipolletta, D.; Cremasco, V.; Crystal, A.; Hewes, B.; Mueller, B.; Quaratino, S.; Sabatos-Peyton, C.; Petruzzelli, L.; et al. Prospects for Combining Targeted and Conventional Cancer Therapy with Immunotherapy. Nat Rev Cancer 2017, 17, 286–301, doi:10.1038/nrc.2017.17.
  33. Luke, J.J.; Flaherty, K.T.; Ribas, A.; Long, G.V. Targeted Agents and Immunotherapies: Optimizing Outcomes in Melanoma. Nat Rev Clin Oncol 2017, 14, 463–482, doi:10.1038/nrclinonc.2017.43.
  34. Gutzmer, R.; Stroyakovskiy, D.; Gogas, H.; Robert, C.; Lewis, K.; Protsenko, S.; Pereira, R.P.; Eigentler, T.; Rutkowski, P.; Demidov, L.; et al. Atezolizumab, Vemurafenib, and Cobimetinib as First-Line Treatment for Unresectable Advanced BRAFV600 Mutation-Positive Melanoma (IMspire150): Primary Analysis of the Randomised, Double-Blind, Placebo-Controlled, Phase 3 Trial. Lancet 2020, 395, 1835–1844, doi:10.1016/S0140-6736(20)30934-X.
  35. Carreau, N.A.; Pavlick, A.C. Nivolumab and Ipilimumab: Immunotherapy for Treatment of Malignant Melanoma. Future Oncol 2019, 15, 349–358, doi:10.2217/fon-2018-0607.
  36. Rosner, S.; Kwong, E.; Shoushtari, A.N.; Friedman, C.F.; Betof, A.S.; Brady, M.S.; Coit, D.G.; Callahan, M.K.; Wolchok, J.D.; Chapman, P.B.; et al. Peripheral Blood Clinical Laboratory Variables Associated with Outcomes Following Combination Nivolumab and Ipilimumab Immunotherapy in Melanoma. Cancer Med 2018, 7, 690–697, doi:10.1002/cam4.1356.
  37. Kozar, I.; Margue, C.; Rothengatter, S.; Haan, C.; Kreis, S. Many Ways to Resistance: How Melanoma Cells Evade Targeted Therapies. Biochim Biophys Acta Rev Cancer 2019, 1871, 313–322, doi:10.1016/j.bbcan.2019.02.002.

Comment 3: In the discussion section, few details are reported about the description of molecular features of each clinical stage. In my opinion, the authors should investigate in details the data about this points. In addition, I would also encouraged the authors to analyze other molecular events that drastically impact on melanoma patients (PD-L1, TMB). How this molecular signature may be infleunced by NGS approach and could play a pivotal role in the decision making process for melanoma patients

The new section 7 includes information about the crucial role that immune checkpoint molecules such as PD-1, PD-L1, and CTLA-4, have in the regulation of the immune system and their link with melanoma immunotherapy. Also, the importance of NGS is highlighted in the context of personalized medicine for melanoma management. The significance of tumor mutational burden (TMB) in melanoma is also highlighted throughout the manuscript, in several different sections (although we didn’t think it was necessary to add a specific section on TMB, since this topic is covered by conspicuous mention to it in many sections).

We believe that the rest of the melanoma molecular features associated with each clinical stage are sufficiently covered, both regarding the traditional histologic classification and the new genetic classification.

Comment 4: Accordingly, I would strongly suggest to investigate in a dedictaed paragraph the emerging molecular biomarkers and the clinical relationship in this patients setting.

We completely agree with the reviewer, and that is the main reason why new section 7 has been included in the manuscript. This section integrates the relationship between molecular biomarkers (molecular targets), target therapies, immunotherapy and cancer acquired drug resistance.

Furthermore, mentions to these new section throughout the text (in the abstract, introduction, methods, etc.) have also been written in red.

Reviewer 2 Report

In the current review authors have well written a brief summary of how UV-induced somatic mutations which leads to clonal evolution leading to melanoma. Overall the review is well written. I have one important suggestion for authors if they can add a section for the current therapies and how mutations affect the resistance to existing therapies and what can be done to address them. This would be a very important section as the therapeutic resistance is higher in cancer than any other disease due to the clonal evolution. 

Author Response

Reviewer 2

According to Reviewer 2, the manuscript should integrate a section about current therapies for melanoma treatment, and the effect that gene mutations have on drug resistance.

We agree with the reviewer on the importance of this issue in the context of melanoma, so the following main section has been added to the manuscript:

  1. New melanoma therapies in in the present-day clinical practice

In this section, we treat in detail the new current targeted therapies that have been developed for melanoma treatment, with emphasis on those that have been approved by the FDA (BRAF inhibitors and MEK inhibitors).

Furthermore, we also deal with the problem of tumor acquired drug resistance to these inhibitors, and its main molecular causes, leading to the appearance of melanoma relapses.

Finally, the new successful immunotherapies are also described in length.

We hope that with the addition of this new lengthy and detailed section we are able to respond to the comments of this referee.

All changes made in the manuscript have been written in red for a convenient detection.

Round 2

Reviewer 1 Report

The manuscript is now suitable for publication without any revisions